# Benefits of FAIMS to Improve the Proteome Coverage of Deteriorated and/or Cross-Linked TMT 10-Plex FFPE Tissue and Plasma-Derived Exosomes Samples

**DOI:** 10.3390/proteomes11040035

**Published:** 2023-10-24

**Authors:** Ana Montero-Calle, María Garranzo-Asensio, Raquel Rejas-González, Jaime Feliu, Marta Mendiola, Alberto Peláez-García, Rodrigo Barderas

**Affiliations:** 1Chronic Disease Programme (UFIEC), Instituto de Salud Carlos III, 28220 Majadahonda, Spain; mgarranzo@isciii.es (M.G.-A.); raquel.rejas@isciii.es (R.R.-G.); 2Translational Oncology Group, La Paz University Hospital (IdiPAZ), 28046 Madrid, Spain; jfeliu.hulp@salud.madrid.org; 3Center for Biomedical Research in the Cancer Network (CIBERONC), Instituto de Salud Carlos III, 28046 Madrid, Spain; marta.mendiola@salud.madrid.org; 4Molecular Pathology and Therapeutic Targets Group, La Paz University Hospital (IdiPAZ), 28046 Madrid, Spain; alberto.pelaez@idipaz.es

**Keywords:** FAIMS, Orbitrap Exploris 480, FFPE tissue, exosomes, protein coverage, proteomics, protein identification and quantification, mass spectrometry

## Abstract

The proteome characterization of complex, deteriorated, or cross-linked protein mixtures as paired clinical FFPE or exosome samples isolated from low plasma volumes (250 µL) might be a challenge. In this work, we aimed at investigating the benefits of FAIMS technology coupled to the Orbitrap Exploris 480 mass spectrometer for the TMT quantitative proteomics analyses of these complex samples in comparison to the analysis of protein extracts from cells, frozen tissue, and exosomes isolated from large volume plasma samples (3 mL). TMT experiments were performed using a two-hour gradient LC-MS/MS with or without FAIMS and two compensation voltages (CV = −45 and CV = −60). In the TMT experiments of cells, frozen tissue, or exosomes isolated from large plasma volumes (3 mL) with FAIMS, a limited increase in the number of identified and quantified proteins accompanied by a decrease in the number of peptides identified and quantified was observed. However, we demonstrated here a noticeable improvement (>100%) in the number of peptide and protein identifications and quantifications for the plasma exosomes isolated from low plasma volumes (250 µL) and FFPE tissue samples in TMT experiments with FAIMS in comparison to the LC-MS/MS analysis without FAIMS. Our results highlight the potential of mass spectrometry analyses with FAIMS to increase the depth into the proteome of complex samples derived from deteriorated, cross-linked samples and/or those where the material was scarce, such as FFPE and plasma-derived exosomes from low plasma volumes (250 µL), which might aid in the characterization of their proteome and proteoforms and in the identification of dysregulated proteins that could be used as biomarkers.

## 1. Introduction

Paired human clinical samples, such as plasma and formalin-fixed paraffin-embedded (FFPE) tissue samples from patients, are valuable specimens that might provide relevant information, from biomarker discovery to the identification of therapeutic targets of intervention. Consequently, these samples should be used in reduced quantities to minimize the loss of material in their analysis by omics approaches. FFPE samples are indeed the most widely used material in clinical routines because they preserve cellular morphology, maintain tissue properties, and allow for long-term storage. These samples are commonly annotated with valuable clinical information. This includes gender, age, clinicopathological data, date of diagnosis, treatment, response to treatment, survival, etc. This allows for retrospective analyses to investigate different biomedical questions [1,2]. In addition, FFPE tissue has been the standard material since the late 1800s for pathological routine diagnostics because it allows for the precise and reproducible separation of local tissue regions [2,3,4]. Indeed, FFPE samples preserve the morphology (shape and structure) of the specimen and its subsequent analysis by histological techniques such as immunohistochemistry. Additionally, nowadays, DNA- and RNA-based analysis are also well-established techniques for FFPE tissue [5,6]. However, proteomics analyses using FFPE samples are yet to be established worldwide because of sample limitations, and protein cross-linking avoids identifying and quantifying as many proteins as if frozen tissue or cell protein extracts were used [1].

Exosomes—extracellular vesicles 30–150 nm in size—are mediators of local and distal intercellular communication via blood or lymphatic vessels [7,8]. Exosomes can be released by normal cells, pathological cells, or cells from the microenvironment [7,8]. For example, exosomes isolated from the plasma of cancer patients have been reported to be enriched in tumor-associated antigens (cancer neoantigens) [9] and are involved in all steps of tumorigenesis, from cancer formation to metastatic colonization [10,11,12,13]. Moreover, brain-derived exosomes have been described as possessing neurotoxic or neurodegenerative effects [14].

In this context, paired exosomes, plasma, and FFPE clinical samples are precious specimens not only for biomarker discovery but also to correlate clinical data with omics findings. Additionally, the extraction of exosomes from minimal plasma samples can be a challenge [9].

Liquid chromatography-tandem mass spectrometry (LC-MS/MS) analysis is the most widely used method for proteome profiling of FFPE tissue, exosomes, and cells by using all major quantitative approaches as tandem mass tags (TMT) or isobaric tags for relative and absolute quantitation (iTRAQ) [1,2,15,16]. These techniques are performed in combination either with data-dependent acquisition (DDA) or with data-independent acquisition (DIA) methods [1,2,15,16]. A challenge facing any proteomics approach is the large number of proteins present in a sample and the large differences in expression between proteins in a tissue, cell, or plasma, which even surpass the throughput of LC-MS/MS systems and reduce the dynamic range of the technique. An important issue in the analysis of clinical samples, including FFPE material, is sample-to-sample carryover, which can negatively affect results. Therefore, simplifying and streamlining proteomics workflows, in particular for clinical applications, would reduce the number of sample processing steps where variations can be found while, at the same time, supporting a higher throughput and increasing the depth into the analyzed proteome.

In recent years, high-field asymmetric waveform ion mobility spectrometry (FAIMS), a variant of ion mobility spectrometry, has generated interest due to its reported utility in increasing the depth of the proteome and protein coverage [17,18,19]. In summary, FAIMS uses an array of two coaxial cylindrical electrodes mounted at the front end of the mass spectrometer where ions are displaced by applying an oscillating asymmetric waveform, the dispersion voltage (DV), to one of the electrodes, which is the amplitude of the waveform. By alternating high and low electric fields between the electrodes, the difference in ion mobility creates a lateral displacement between the electrodes, and a direct current compensation voltage (CV) applied to the inner electrode dictates the ions that are filtered through the mass spectrometer. FAIMS works on a time scale of milliseconds, and ion filtering is based on the state, shape, conformation, and size of ions in the gas phase, whereas singly charged ions are driven out. The CV changes the subset of ions transferred to the mass spectrometer, filtering out non-optimized ion species, such as single-charged ions. Recent studies have demonstrated the potential advantages of coupling FAIMS Pro Duo Interface to mass spectrometers for LC-MS/MS -based proteomics applications [18,20]. However, the benefits of using FAIMS are still a matter of debate [21], as fewer peptides are obtained while maintaining or increasing the number of proteins identified and quantified [21,22]. In fact, very few laboratories routinely use FAIMS on their LC-MS/MS equipment.

Accordingly, our aim here was to investigate the applicability of FAIMS Pro Duo Interface coupled to an Orbitrap Exploris 480 to increase the depth into the proteome of paired FFPE tissue samples and exosomes isolated from 250 µL plasma samples from colorectal cancer patients with or without *KRAS* mutations, in comparison to protein extracts derived from cancer cell cultures, frozen non-cross-linked tissues, and exosomes isolated from large plasma volumes (3 mL), while providing specific CV methods for FAIMS analysis. To this end, we analyzed a combination of two or three CVs at 15 V intervals with FFPE and paired exosome protein samples in comparison to cell protein extracts. FAIMS results showed improved protein coverage, number of peptides (>70%), and number of proteins (>100%) in FFPE and exosome samples isolated from 250 µL of plasma. Additionally, limited improvements up to 26% were found in the number of identified and quantified proteins with FAIMS when analyzing protein extracts from cancer cells, frozen tissue samples, or exosome samples isolated from large plasma volumes, demonstrating the noticeable usefulness of FAIMS to increase the depth into the proteome over control experiments without FAIMS in selected scarce, cross-linked, and/or deteriorated biological samples.

## 2. Materials and Methods

### 2.1. Human Samples

Paired plasma and FFPE tissue samples from colorectal cancer (CRC) patients were obtained from the Hospital La Paz (IdiPAZ, Madrid, Spain) biobank, which belongs to the National Biobank Network (ISCIII) cofounded with FEDER funds. The Institutional Ethical Review Boards of the Instituto de Salud Carlos III approved this study (CEI PI 13_2020-v2, and CEI PI 49). Tissue samples were collected using a standardized sample collection protocol, histopathologically analyzed, and stored at 4 °C until use [23,24,25].

Ten paired FFPE and plasma samples (250 µL) from CRC patients at stage III from the IdIPAZ biobank were used in this study (Appendix A). Six patients’ tumors possessed *KRAS* mutations at codon 12 or 13 at exon 2, and four presented wild-type *KRAS.* The complete information of CRC patients regarding gender, age, race, ethnicity, and *KRAS* status is depicted in Appendix A. CRC patients were assessed for eligibility according to the following inclusion criteria. Male and female CRC patients who: (1) underwent surgery to remove a CRC tumor in stage III with or without *KRAS* mutation; (2) gave a blood sample for plasma collection before surgery for scientific investigation; and (3) had enough paired FFPE tumor samples for scientific investigations. Paired plasma samples and FFPE tissues were obtained from CRC patients between 2015 and 2018. Additionally, ten non-paired plasma samples from CRC patients at stages I (n = 2) and IV (n = 2) of the disease, individuals with premalignant lesions (adenomas, n = 2), and healthy individuals (n = 3) from the IdISSC biobank were used for the isolation of exosomes from a large plasma volume (3 mL) (Appendix A). Written informed consent was obtained from all patients.

Frozen tissue samples from the left prefrontal cortex of Alzheimer’s disease (AD) patients (n = 12), patients with vascular (VD) (n = 2), frontotemporal dementia (FTD) (n = 5), and healthy individuals (n = 2) from the CIEN Foundation’s Tissue Bank (BT-CIEN) were used for mass spectrometry analyses [26]. Samples were processed and classified following standard protocols, as previously described [26,27].

### 2.2. CRC Cells and Transfection

The isogenic KM12 CRC cell model, composed of the low-metastatic KM12C cells and the high-metastatic liver KM12SM cells, was obtained from I. Fidler’s laboratory (MD Anderson Cancer Center, Houston, TX, USA). CRC cells were grown at 37 °C and 5% CO_2_ in Dulbecco’s Modified Eagle Medium (DMEM, Lonza, Basel, Switzerland) containing 10% fetal bovine serum (FBS, Sigma-Aldrich, St. Louis, MI, USA), 1× L-glutamine (Lonza), and 1× penicillin/streptomycin (Lonza) (complete medium).

For SPRYD7 overexpression, CRC cells were transfected with the JetPrime transfection reagent (Polyplus, Illkirch-Graffenstaden, France) [28,29]. Prior to transfection, the *SPRYD7* gene was cloned into the pcDNA3.1(+) expression vector (Thermo Fisher Scientific, Waltham, MA, USA) with the NEBuilder HiFi DNA assembly method, following the manufacturer’s instructions [29]. The sequence was verified prior to use. Then, 2.5 × 10^5^ cells seeded on a 6-well plate (corning) in 2 mL of complete DMEM medium were transfected. Briefly, 2 µg of SPRYD7 (SPRYD7-stably transfected cells) or empty (mock-stably transfected cells) pcDNA3.1(+) vectors were diluted in 200 µL of JetPRIME buffer and incubated with 4 µL of JetPRIME Transfection reagent for 10 min at room temperature (RT). Then, the reaction solution was added to the cells, and the cells were incubated at 37 °C and 5% CO_2_ for 48 h. Next, complete medium was removed, and KM12 cells were grown in complete medium containing 1 mg/mL G418 (Geneticin 418, Santa Cruz BioTechnology, Dallas, TX, USA) for selection during 3–4 weeks. Finally, positively selected transfected cells were grown in a complete medium supplemented with 0.6 mg/mL G418 to establish genetically modified CRC cell lines.

### 2.3. Plasma Exosome Isolation and Purification

Exosomes were alternatively isolated from FFPE-paired 250 µL plasma samples from CRC patients at stage III, or from non-paired 3 mL plasma samples from CRC patients at stages I or IV, individuals with premalignant lesions, or from healthy individuals by differential centrifugation as previously described [9,30]. Briefly, plasma samples were centrifuged at 20,000× *g* for 15 min at 4 °C. Then, supernatants were centrifuged at 10,000× *g* for 30 min in a Beckman-Coulter (Brea, CA, USA) ultracentrifuge XL-100 K to remove microvesicles, and subsequently, supernatants were centrifuged at 100,000× *g* for 70 min at 4 °C to sediment exosomes. Finally, supernatants were discarded, and exosomes were washed with PBS and subsequently centrifuged at 100,000× *g* for 70 min at 4 °C. Finally, exosomes were resuspended in 500 µL of PBS and stored at −80 °C until use.

### 2.4. Transmission Electron Microscopy

The size of the isolated extracellular vesicles was analyzed by transmission electron microscopy (TEM) with negative staining [9,30]. Furthermore, 5 µL of each exosome sample were fixed for 5 min in 2% PFA in PBS 1× and incubated over glow-discharged carbon-coated grids for 5 min. Then exosomes were negatively stained with 2% aqueous uranyl acetate, and samples were analyzed on a FEI Tecnai 12 electron microscope (FEI company, Hillsboro, OR, USA) equipped with a LaB6 filament operated at 120 kV. All images were recorded with an FEI Ceta digital camera.

### 2.5. Protein Extraction and Quantification

Deparaffinization of FFPE tissue samples from stage III CRC patients was performed prior to protein extraction. Tissue samples were washed twice with 500 µL heptane (Sigma-Aldrich) for 1 h at RT and 700 rpm after 10 s of mixing in a vortex. Then, 25 µL of 100% methanol was added to each sample for tissue rehydration. After vigorous agitation on a vortex for 10 s, samples were centrifuged at 15,000× *g* for 5 min to collect tissues. Next, deparaffinized tissue was lysed with 300 µL of lysis buffer (RIPA buffer, Sigma-Aldrich) supplemented with 1× protease and phosphatase inhibitors (MedChemExpress, Monmouth Junction, NJ, USA) by mechanical disaggregation using the TissueLyser II (Qiagen, Hilden, Germany) (2 cycles of 30 s at 30 Hz). Tissues were then incubated at 700 rpm and 100 °C for 20 min, and subsequently at 80 °C for 2 h to remove paraffin traces and reverse crosslinking. Finally, samples were centrifuged at 10,000× *g* and 4 °C for 10 min, and protein extracts (supernatants) were transferred to a new tube and stored at −80 °C until use.

Protein extracts from CRC cells were obtained after cell detachment at 90% confluence with PBS 1× containing 4 mM EDTA (Carl Roth, Karlsruhe, Baden-Wurttemberg, Germany). Then, cells were centrifuged at 260× *g* at RT for 5 min, supernatants discarded, and cell pellets were manually lysed with 500 µL of RIPA buffer supplemented with 1× protease and phosphatase inhibitors using 16 G and 18 G needle syringes. Samples were then centrifuged at 10,000× *g* and 4 °C for 10 min, and protein extracts (supernatants) were collected and stored at −80 °C until use.

Tissue and cell protein extracts were quantified by the tryptophan quantification method [31,32], whereas the protein concentration of exosome samples was obtained using the MicroBCA Protein Assay Kit (Thermo Fisher Scientific). Protein extract concentrations were finally confirmed by Coomassie blue staining and western blot (WB) after 10% SDS-PAGE separation.

### 2.6. Western Blot

Ten micrograms of each protein extract from CRC patients’ tissues and cells were separated on 10% SDS-PAGE under reducing conditions. Alternatively, exosome samples were lysed with loading buffer supplemented with 1.5% β-mercaptoethanol (five cycles of 5 min on ice and 5 min at 95 °C) prior to protein separation in SDS-PAGE. Proteins were then transferred to nitrocellulose membranes at 100 V for 90 min and incubated with the mouse monoclonal anti-Alix (Santa Cruz Biotechnology, sc-53540) or the mouse monoclonal anti-CD63 (HansaBioMed, Tallinn, Estonia, HBM-CD63-xx) antibodies 1:500 diluted in blocking buffer (0.1% Tween PBS 1× supplemented with 3% skimmed milk) overnight (O/N) at 4 °C and in rotation after blocking for 1 h at RT. Then, membranes were washed three times with washing buffer (0.1% Tween PBS 1×) and incubated with HRP-anti-mouse IgG (Sigma-Aldrich, A4416) 1:1000 diluted in blocking buffer for 1 h at RT and in rotation.

### 2.7. RNA Extraction, cDNA Synthesis, and PCR

For RNA extraction, CRC cells were grown until 90% confluence and detached with trypsin-EDTA (Lonza). After centrifugation at 260× *g* at RT for 5 min, supernatants were discarded, and cell pellets were resuspended in 500 µL of NZYol (NZYTech, Lisbon, Portugal) and incubated for 5 min at RT. Then, the solution was further incubated with 100 µL of chloroform (Sigma-Aldrich) for 3 min at RT after mixing by inversion, and centrifuged at 12,000× *g* for 15 min and 4 °C. Subsequently, the upper phase with the isolated RNA was transferred to a new tube, mixed with 1× volume 70% ethanol, and purified using the RNeasy Mini Kit (Qiagen) following the manufacturer’s instructions. Finally, RNA was eluted in 100 µL of DEPC water and quantified with the Nanodrop One (Thermo Fisher Scientific).

cDNA was synthesized from 1 µg of RNA using the NZY First-Strand cDNA Synthesis Kit (NZYtech), according to the manufacturer’s instructions. Then, per PCR for the analysis of SPRYD7 overexpression using specific oligonucleotides (Fw: GTCCAGCATCAGGTATACGAGG, Rv: CAAAACCAGGTGGAGGCGTATG), 0.8 µL of cDNA was used. GAPDH was amplified using specific oligonucleotides (Fw: GTCTCCTCTGACTTCAACAGCG, Rv: ACCACCCTGTTGCTGTAGCCAA) as a loading control.

### 2.8. 10-Plex TMT Labeling

For the TMT analyses, individual cell protein extracts, plasma exosomes, and FFPE tissue samples were used, whereas pooled samples were used for the TMT experiment with frozen left prefrontal brain tissue samples (Healthy, FTD, Braak V, and Braak VI pools were used in duplicate). Furthermore, 10 μg of each protein extract in 100 μL of RIPA were reduced with 10 μL of 100 mM TCEP for 45 min at 37 °C and 600 rpm and alkylated with 11 μL of 400 mM chloroacetamide for 30 min at RT, 600 rpm, and in darkness. Prior to reduction and alkylation, exosome samples were lysed in RIPA buffer (five cycles of 5 min on ice and 5 min at 95 °C). In addition, for the TMT analysis of cell protein extracts, KM12C mock-stably transfected and SPRYD7 overexpressing cells were labeled in triplicate, whereas KM12SM cells were labeled in duplicate. Furthermore, CRC cells were incubated for 48 h in DMEM-free FBS medium at 95% confluence to remove any trace of bovine serum albumin (BSA) from the samples.

Then, protein extracts were incubated with 100 µL of SeraMag magnetic beads mix (50% hydrophilic beads–50% hydrophobic beads, GE Healthcare, Chicago, IL, USA) and 200 μL of acetonitrile (ACN) for 35 min at RT and 600 rpm for protein binding to the beads. Then, supernatants were discarded, and magnetic beads were washed twice with 70% ethanol and once with ACN. Finally, supernatants were discarded, and proteins were O/N digested at 37 °C and 600 rpm with 0.5 μg of porcine trypsin (Thermo Fisher Scientific) in 100 μL of 200 mM HEPES, pH 8.0. The day after, samples were sonicated twice, supernatants collected, and separately labeled with the ten different Tandem Mass Tags reagents (Thermo Fisher Scientific) in two incubation steps of 30 min at RT and 600 rpm and with 10 μL of reagent per incubation. Finally, samples were incubated with 10 μL of 1 M glycine, pH 2.7, for 30 min at RT and 600 rpm. Next, the contents of the 10 tubes were pooled together and dried under vacuum prior to peptide separation using the High-pH Reversed-Phase Peptide Fractionation Kit (Thermo Fisher Scientific). Briefly, desiccated peptides were reconstituted in 300 μL of 0.1% TFA in H_2_O_mq_, and columns were equilibrated twice with 300 μL of ACN and twice with 300 μL of 0.1% TFA in H_2_O_mq_. Then, peptides were loaded into the columns, washed twice with 300 μL of 0.1% TFA in H_2_O_mq_, and separated in 12 fractions of 300 µL each in 0.1% triethylamine and 2.5–100% ACN. Fractions were then mixed in six fractions by pooling the latest fractions with the initial ones (2, 9, and 1; 7 and 3; 10 and 4; 8 and 5; 11 and 6; and 12), dried under vacuum, and stored at −80 °C until analysis in six LC-MS/MS runs. Samples were reconstituted in 10 μL of 0.1% FA prior to their injection onto the LC-MS/MS mass spectrometer.

### 2.9. LC-MS/MS Analysis

TMT experiments were analyzed on an Orbitrap Exploris 480 mass spectrometer (Thermo Fisher Scientific) equipped (or not) with the FAIMS Pro Duo interface (Thermo Fisher Scientific). Peptide separation was performed on the Vanquish Neo UHPLC System (Thermo Fisher Scientific). For each analysis, samples were loaded into a precolumn PepMap 100 C18 3 µm, 75 µm × 2 cm Nanoviper Trap 1200BA (Thermo Fisher Scientific) and eluted in an Easy-Spray PepMap RSLC C18 2 µm, 75 µm × 50 cm (Thermo Fisher Scientific) heated at 50 °C. The mobile phase flow rate was 300 nL/min, and 0.1% FA in H_2_O_mq_ and 0.1% FA in 80% ACN were used as buffers A and B, respectively. The 2 h gradient was: 0–2% buffer B for 4 min, 2% buffer B for 2 min, 2–42% buffer B for 100 min, 42–72% buffer B for 14 min, 72–95% buffer B for 5 min, and 95% buffer B for 10 min. Samples were re-suspended in 10 µL of buffer A, and 2–4 µL (800 ng) of each sample were injected per run. For ionization, 1900 V of liquid junction voltage and 280°C capillary temperature were used. The full scan method employed a *m*/*z* 350–1400 mass selection, an Orbitrap resolution of 60,000 (at *m*/*z* 200), an automatic gain control (AGC) value of 300%, and a maximum injection time (IT) of 25 ms. After the survey scan, the 12 most intense precursor ions were selected for MS/MS fragmentation. Fragmentation was performed with a normalized collision energy of 34, and MS/MS scans were acquired with a 100 *m*/*z* first mass, an AGC target of 100%, a resolution of 15,000 (at *m*/*z* 200), an intensity threshold of 2 × 10^4^, an isolation window of 0.7 *m*/*z* units, a maximum IT of 22 ms, and the TurboTMT enabled. Charge state screening was enabled to reject unassigned, singly charged, and greater than or equal to seven protonated ions. A dynamic exclusion time of 30 s was used to discriminate against previously selected ions. For FAIMS, a gas flow of 4.7 L/min and CVs = −45 V and −60 V, or CVs = −30 V, −45 V, and −60 V, were used.

### 2.10. Data Analysis and Statistical Analysis

MS data were analyzed with MaxQuant (version 2.1.3, Max Planck Institute of Biochemistry, Planegg, Germany) using standardized workflows. Mass spectra *.raw files were searched against the Uniprot UP000005640_9606.fasta *Homo sapiens* (human) 2022 database (20,577 protein entries) using reporter ion MS2 type for TMTs. Precursor and reporter mass tolerances were set to 4.5 ppm and 0.003 Da, respectively, allowing 2 missed cleavages. Carbamidomethylation of cysteines was set as a fixed modification, and methionine oxidation, N-terminal acetylation, and Ser, Thr, and Tyr phosphorylation were set as variable modifications. Unique and razor peptides were considered for quantification. Minimal peptide length and maximal peptide mass were fixed to 7 amino acids and 4600 Da, respectively. Identified peptides were filtered by their precursor intensity fraction (PIF) with a false discovery rate (FDR) threshold of 0.01. Proteins identified with at least one unique peptide and an ion score above 99% were considered for evaluation, whereas proteins identified as potential contaminants were excluded from the analysis. The protein sequence coverage was estimated for specific proteins by the percentage of matching amino acids from the identified peptides having confidence greater than or equal to 95% divided by the total number of amino acids in the sequence. In addition, reporter ion intensities were bias corrected for the overlapping isotope contributions from the TMT tags according to the manufacturer’s certificate.

Next, data normalization was performed to equalize the differences in the total sum of signals for each TMT channel, as the same amount of protein was labeled in each TMT sample. Sample loading (SL) normalization was performed with R Studio (version 4.1.1, Posit PBC, Boston, MA, USA) according to the established protocol (https://github.com/pwilmart, accessed on 2 November 2022), using the “tidyverse”, “psych”, “gridExtra”, “scales”, and “ggplot2” packages. For the comparison between paired tissue and plasma exosome samples, the two independent TMT analyses were normalized using the IRS (internal reference scaling) normalization according to the established protocol (https://pwilmart.github.io/IRS_normalization/understanding_IRS.html, accessed on 2 November 2022).

For statistical analysis, due to the low number of samples per group and the high number of variables, an empirical Bayes-moderated t-statistics analysis was performed with R Studio (version 4.1.1) using the packages “limma”, “dplyr”, “tidyverse”, “ggplot2”, and “rstatix”, according to previously described procedures [33,34,35,36]. This method is a procedure for statistical interference that estimates the probability distribution from the data. By calculating a trend line on protein means versus variances, a new variance is interpolated for each individual protein measured, and then the mean protein expression values of each group of replicates for all the identified and quantified proteins can be compared. Correction for multiple tests was not performed to not increase the number of type-II errors (false negatives) and lose any potential dysregulated proteins associated with SPRYD7 overexpression or *KRAS* mutations. Prior to statistical analysis, reverse and contaminant proteins were removed, and data filtering (proteins identified in at least 30% of samples were considered for the analysis) and missing value imputation by random draws from a Gaussian using the “imputeLCMD” R package were performed. Proteins identified with one or more unique peptides, an expression ratio ≥ 1.5 (upregulated) or ≤0.67 (downregulated), and a *p*-value ≤ 0.05 were selected as statistically significant dysregulated proteins. Expression ratio cut-offs were selected according to previous reports [26,29,30,31,37]. Venn diagrams were obtained using the Jvenn website (http://bioinfo.genotoul.fr/jvenn, accessed on 21 May 2023). Pearson correlation analyses were performed with the “Stats” package of R Studio (version 4.1.1).

## 3. Results

### 3.1. Proteomics Analysis of Paired FFPE and Exosome Protein Extracts for the Identification of Dysregulated Proteins Involved in Colorectal Cancer

Colorectal cancer is the third most common cancer type and the second cause of cancer-related death worldwide, mainly due to liver metastasis [38]. Currently, CRC diagnosis is based on the identification of fecal occult blood, which is not specific enough to the disease, and colonoscopy, which is an invasive technique that requires previous sedation and bowel preparation. For that reason, the identification of novel dysregulated proteins associated with CRC, which could be used as biomarkers of the disease and allow for an in-depth analysis of its pathogenesis, is crucial. In this context, proteomics techniques have been widely used for the characterization of the proteome associated with CRC [31,39,40,41,42], which might aid in a better understanding of the biology of a disease and in the identification of dysregulated proteins with potential as diagnostic or prognostic biomarkers. One of the most common proteomics approaches for the identification of dysregulated proteins associated with a pathology is TMT, which allows for the simultaneous analysis of up to 18 samples (with the potential to evaluate 6, 10, 11, 16, or 18 samples simultaneously) in a single experiment by the covalent binding of reporters with a differential mass to the peptides from each sample. Additionally, TMT allows for the analysis of any protein extract from different biological sources, such as human tissue and plasma samples or cell cultures, which is mandatory for a further understanding of a disease, as each biological source provides different and valuable information about the pathology. Since the complexity of each biological source is different, each source might need to be analyzed differently with the objective of achieving as much information as possible that could be useful for the study of the pathology.

In this context, we aimed to explore in this work the benefits of FAIMS Pro Duo Interface on an Orbitrap Exploris 480 to improve peptide and protein identifications and quantifications using protein extracts from different biological sources: hard-to-analyze paired plasma exosomes isolated from 250 µL of plasma and FFPE tissue samples from CRC patients from deteriorated, cross-linked, or in small quantities clinical samples, and easy-to-work CRC cell protein extracts derived from immortalized cell sources. To this end, we selected paired plasma and FFPE tissues from mismatch repair proficient CRC tumors with *KRAS* mutations at codon 12 or 13, exon 2 (Mut), or *KRAS* gene wild-type (WT), from which we were unable to get accurate results using a Q Exactive Orbitrap mass spectrometer, and CRC cells (low metastatic KM12C and high metastatic to liver KM12SM CRC cells) stably overexpressing SPRYD7, a gene previously described as involved in CRC carcinogenesis [31,43], to identify proteins dysregulated in CRC and in CRC patients associated with these alterations (*KRAS* mutation and SPRYD7 overexpression). Regarding exosomes, paired plasma exosomes isolated by differential centrifugation were also investigated to identify dysregulated exosome proteins involved in cell-cell communication and carcinogenesis associated with mutated *KRAS*. After trypsin digestion and TMT labeling, all samples from each experiment were pooled together and peptides separated in 12 fractions according to their hydrophobicity using an ACN gradient in 0.1% triethylamine prior to LC-MS/MS analysis. These 12 fractions were then pooled together into six fractions, which were subsequently analyzed on an Orbitrap Exploris 480 equipped or not with the FAIMS Pro Duo Interface. A scheme of the workflow of the study is depicted in Figure 1.

### 3.2. LC-MS/MS Analysis of Protein Samples

Prior to proteomics sample processing, the quality of cell, tissue, and plasma exosome protein extracts was confirmed by Coomassie blue staining (Figure 2A–C). Importantly, although 10 µg of protein extracts were loaded per line, several differences were observed among the samples. The cell protein extracts showed a similar protein distribution across the lines (Figure 2A). In contrast, FFPE-derived protein extracts showed cross-linking because of the formalin fixation, and proteins did not enter the separation part of the gel due to the high molecular weight of the cross-linked proteins. Thus, just a smearing along the line was observed (Figure 2B), which corresponded to deteriorated or fragmented cross-linked proteins. Finally, the protein content derived from the exosomes showed three dominant bands around 55 kDa and 70 to 100 kDa, which are usually present in exosome lysates and encompass about 20–30% of their protein content, thus reducing the dynamic range of the mass spectrometry analyses (Figure 2C) [9]. Despite these differences, the protein extracts derived from the different sources were similar to those obtained in other studies involving these types of biological samples [9,30,44]. Moreover, the overexpression of SPRYD7 in KM12C and KM12SM CRC cells was confirmed by PCR analysis (Figure 2D). Additionally, plasma exosomes were characterized by TEM and WB to analyze the compatibility of the extracellular vesicles isolated and purified by differential centrifugation with exosome vesicles. Extracellular vesicles lower than 100 nm were observed by TEM in all samples (Figure 2E). Furthermore, two proteins specific to extracellular vesicles, Alix and CD63, were observed in all samples by western blot (Figure 2F), confirming the quality of the exosome samples for the proteomics analyses. Finally, the protein concentration of each cell, tissue, or exosome protein extract was adjusted according to the intensity of the total protein content per well regarding the Coomassie blue staining to ensure that similar protein amounts were used for each sample in the three TMT analyses.

Then, three 10-plex TMT quantitative proteomics experiments were performed and analyzed using the Orbitrap Exploris 480 with and without FAIMS to compare the performance and determine whether FAIMS could produce an increase in peptide and protein identifications and quantifications. Experiments were performed under the same conditions. In brief, six fractions were run per TMT experiment. We injected 2 µL of sample (800 ng peptides) per run. A two-hour ACN gradient was used for mass spectrometry analysis. In addition, we used two CVs (CV = −45 V and CV = −60 V) with the FAIMS Pro Duo Interface. Prior to further analysis, proteomics data from the three TMTs with or without FAIMS were normalized. Interestingly, although small differences intra- and inter-TMT were observed among the TMT relative intensities (Appendix A), the results suggest that similar amounts of protein were used per channel after adjusting the protein concentration according to Coomassie blue staining and WB analysis. Since in TMT experiments peptide quantification is mainly dependent on the labeling step, we first investigated the number of missing values for peptides and proteins to analyze the efficiency of the TMT labeling performed (Figure 3A). Missing values for peptides and proteins were lower than 1% and 15%, respectively, in the three experiments, confirming the quality of the proteomics data. In addition, the number of missing values was lower in two out of the three TMT experiments (cells and FFPE tissues) when using the FAIMS Pro Duo Interface. Regarding the number of identified and quantified peptides, a 10% reduction was observed for the cell TMT experiment with FAIMS, whereas a significant increase in the number of peptides, greater than 70%, was observed for the FFPE tissues and plasma exosomes experiments with FAIMS (Figure 3B). In concordance with these data, an increase higher than 100% was observed in the number of identified and quantified proteins for the FFPE tissues and exosome experiments with FAIMS, which was lower than 11% for the experiment with cell protein extracts (Figure 3B).

Peptide fractionation prior to LC-MS/MS is crucial to reduce peptide overlapping during mass spectrometry analyses, thus increasing the number of identified peptides and proteins. Here, to reduce the time and cost of TMT analyses, peptide fractions with high differences in their hydrophobic properties (eluted at different ACN percentages) were pooled together, which ensured the accurate separation of peptides during the liquid chromatography fractionation prior to mass spectrometry analysis, as most peptides were only identified in one or two fractions (Figure 4A–F). In addition, we observed that most of the peptides eluted between 7.5 and 50% ACN with or without FAIMS.

FAIMS Pro Duo Interface acts as an ion selection filter by introducing a second peptide separation at the entrance of the mass spectrometer. Therefore, using FAIMS technology with two different CVs, each peptide fraction was subsequently subdivided into two fractions, allowing for better peptide separation without increasing the time consumed per TMT experiment and reducing peptide overlapping and masking. In our experiments, this improvement was demonstrated by a significant increase (higher than 50%) in the number of total peptides and specific peptides per fraction for the FFPE tissue and plasma exosome samples (Figure 4C–F). This was also shown in the significant increase in the total number of specific peptides per fraction combining the two CV fractions in comparison with the results without FAIMS. On the contrary, for the cell protein extracts experiment, a slight decrease in the total number of peptides and specific peptides per fraction was observed with FAIMS technology. The use of FAIMS did not provide significant improvements in peptide identification with the cell protein extracts because the percentage of peptide overlapping and masking without FAIMS was lower in the cell protein samples than in the paired FFPE tissue or exosomes isolated from 250 µL of plasma samples (Figure 4). In contrast, the increased time of each mass spectrometry cycle and the CV filtering when using FAIMS resulted in a reduction in the number of identified peptides in the cell experiment (Figure 4A,B), while allowing an increased peptide fractionation and separation of co-isolating peptides in paired FFPE tissue and plasma exosome samples. However, this decrease in the number of peptides did not result in a reduction in the number of protein identifications, as previously shown (Figure 3B), but in a decrease in the protein coverage (Figure 5A,B).

A 16% decrease, approximately, in the number of proteins identified with more than 20% of protein coverage was observed for the experiment with cell protein extracts with FAIMS (Figure 5A,B). However, for the FFPE tissues and plasma exosomes experiments, we did not observe a reduction in the protein coverage with the FAIMS Pro Duo Interface, as the significant increase in peptide and protein identifications compensated for the losses associated with FAIMS technology previously observed with cell samples (Figure 5C–F). Finally, we could not observe significant differences in the percentage of unique and razor peptides per protein with or without FAIMS in any experiment (Figure 5A–F). Moreover, the increase in the number of peptide identifications was associated with an increase in the number of both unique and razor peptides per protein in the TMT experiments of hard-to-analyze samples.

### 3.3. FAIMS Analyses with Two or Three CVs

Next, we analyzed the same FFPE tissues and plasma exosomes samples in a three-hour ACN gradient using three CVs (CV = −30, CV = −45, and CV = −60) with FAIMS technology. A slight increment in the number of peptide and protein identifications (1–5% approximately) was observed for the plasma exosomes analysis with three CVs in comparison to the two CVs data, whereas a 20% decrease in the number of identifications was observed for the FFPE tissues analyzed (Figure 6A). In addition, with the three CVs, we could also observe a high peptide separation per fraction, not accompanied by an increase in the number of specific peptides per fraction, by combining the three CV fractions in comparison to the experiment with the two CVs (Figure 4D,F and Figure 6B). Finally, we could neither observe significant differences in the protein coverage nor in the number of total peptides per protein between TMT analyses with two and three CVs (Figure 5D,F and Figure 6C). In addition, and according to previous analyses [18], a higher number of peptides were identified at CV = −45 V with two and three CVs, whereas this number significantly decreased at CV = −30 V. Differences lower than 20% in the identification of peptides were observed comparing two CVs (Figure 4D,F) and three CVs (Figure 6). In FFPE-derived protein extracts, 2501 proteins were identified and quantified with two CVs and 2108 with three CVs, whereas for exosomes derived from plasma, 559 and 604 proteins were identified and quantified with two and three CVs, respectively. Therefore, these results, together with the increased consumption time of the mass spectrometer with three CVs (three hours per run), suggest that the use of three CVs should not be useful enough for routine analysis of paired FFPE tissue and plasma exosomes from clinical samples. Additionally, the increase in complexity of the mass spectrometry analysis with three CVs does not improve the quality of the proteomics data.

All these data suggest the benefits of FAIMS Pro Duo Interface to improve the mass spectrometry proteomics analysis of samples derived from deteriorated, cross-linked, and/or samples where the material was scarce, such as paired FFPE and plasma-derived exosomes isolated from low (250 µL) plasma volumes. Advantages were observed in terms of peptide and protein identification, reduced precursor peptide co-isolation, and increased sensitivity. In contrast, mass spectrometry analysis of robust or non-fragmented protein extracts, such as those derived from cancer cells, did not show significant improvements with FAIMS technology (10% improvement, approximately). Our results also suggest that the use of two CVs should be enough for a significant improvement in the proteomics analyses since the 1 h increase in the time of use of the Orbitrap Exploris 480 considering the analysis with three CVs in comparison to the 2 h run when using two CVs does not improve the proteomics analysis of clinical samples. Thus, the use of three CVs was not justified with these clinical samples.

### 3.4. Improvement in the Identification of Dysregulated Proteins Due to FAIMS

As above indicated, the use of FAIMS Pro Duo Interface in mass spectrometry proteomics analyses increased the number of peptide and protein identifications, making this improvement much more beneficial for FFPE tissues and plasma exosomes than for cell protein extracts. In addition, we confirmed that the installation of FAIMS on mass spectrometers did not modify the identified proteins. For the three LC-MS/MS TMT experiments, more than 80% of the proteins identified in the analyses without FAIMS were also identified in the analyses with FAIMS, both with two and three CVs (Figure 7A). In contrast, FAIMS produced a 65–70% increase in the number of novel proteins identified in all cases, whereas approximately 70% of the proteins identified with two CVs were also identified with three CVs, highlighting a high correlation between the two analyses with FAIMS (Figure 7A).

Thus, the FAIMS interface would allow, with the three analyzed protein extracts, in-depth proteome coverage associated with CRC, regardless of their biological source. Therefore, to address this question, after the normalization of mass spectrometry data (Appendix A), a statistical analysis with the R program was performed to determine the number of dysregulated proteins associated with CRC from the three TMT experiments, with or without FAIMS technology (Figure 7B,C). Although slight differences were previously observed in the number of identified and quantified proteins for the experiment with the cell protein extracts with and without FAIMS, an increase in the number of upregulated and downregulated proteins was found with the FAIMS Pro Duo Interface, which was more noticeable for the paired FFPE tissue and plasma exosome experiments.

Finally, using the higher number of protein identifications obtained with FAIMS Pro Duo Interface and two CVs for the paired FFPE tissue and plasma exosome TMT analyses, we investigated the correlation between both paired protein extracts from CRC patients to determine whether the proteins identified and quantified in FFPE cancer tissue and plasma exosomes could become minimally invasive biomarkers. First, we observed a significant difference in the number of identified proteins from tissues and from plasma exosomes due to the different nature of the protein samples (Appendix A). In all cases, about 40–45% of proteins identified from exosome samples were also identified in their corresponding paired tissue samples, which represents approximately 8–10% of the total of proteins identified from tissue samples. Second, the Pearson correlation was calculated with the 231–246 common proteins in the paired FFPE tissue and plasma exosome samples. In all cases, a high positive correlation was observed between paired samples (R > 0.93), but for the *KRAS* WT 4 and the *KRAS* Mut 6 samples, the correlation decreased to R = 0.82 and R = 0.9, respectively (Figure 8). These results suggest that those proteins identified as dysregulated in cancer tissue samples might also be dysregulated in plasma samples, and that their measurement in plasma or in plasma exosomes could become blood-based biomarkers of the disease.

Collectively, our results highlight the benefits of FAIMS Pro Duo Interface to get a deeper characterization of the proteome in hard-to-analyze clinical samples, such as those deteriorated, fragmented, and/or cross-linked protein extracts investigated here from FFPE tissue or exosomes isolated from low volumes of plasma samples.

### 3.5. Role of FAIMS in the MS Analysis of Non-Cross-Linked or Non-Deteriorated Protein Samples

Finally, to confirm the advantages of the FAIMS Pro Duo Interface in the mass spectrometry analyses of protein samples from paired deteriorated, cross-linked, and/or in small quantities clinical samples, we analyzed by TMT 10-plex quantitative proteomics protein extracts from fresh frozen tissue samples and exosomes isolated from large plasma volumes (3 mL) with and without FAIMS (Appendix A).

After data normalization (Appendix A), the quality of the TMT labeling was confirmed by the low number of missing values obtained from identified peptides and proteins (<1% and <15%, respectively), as observed in the previous TMT experiments analyzed. In addition, lower missing values were obtained with the FAIMS technology in the two TMT experiments (Figure 9A). Next, the higher quality of the protein extracts from frozen tissue and high plasma volume exosome samples in comparison to the previous biological samples was noticed by a significant increase in the number of peptides and proteins identified in contrast to cross-linked and/or deteriorated protein samples (Figure 9B). Interestingly, a 12% and 3% reduction in the number of identified and quantified peptides was observed with the FAIMS Pro Duo Technology from frozen tissue samples and exosomes isolated from large (3 mL) plasma volumes, respectively. However, FAIMS allowed for an 11% and 26% increase in the number of identified and quantified proteins, respectively (Figure 9B). These results suggest that the use of the FAIMS interface offers limited improvements for the proteomics analysis of non-cross-linked and non-deteriorated protein samples, as previously observed for cell protein extracts. Additionally, we observed that the analysis of exosomes isolated from large plasma volumes (3 mL) significantly increased the performance of the proteomics analyses with and without the FAIMS Pro Duo interface. This observation would be due to the increased number of exosome particles isolated by centrifugation and to the reduction of the sample ratio between the highly abundant plasma contaminant proteins or highly abundant proteins and the exosome protein content. In this sense, a higher increase in the number of identified and quantified proteins with FAIMS was observed for plasma exosomes isolated from 3 mL of plasma in contrast to frozen tissue or cell protein extracts, suggesting that the FAIMS Pro Duo interface allows for a significant increase in the dynamic range of complex samples despite the presence of highly abundant contaminant proteins, such as plasma proteins (Appendix A).

Similar to the performance observed with the cell extract proteomics analysis, limited differences were observed in the peptide identification, overlapping, and masking with and without the FAIMS Pro Duo interface in the frozen tissue and exosomes isolated from large plasma volume (3 mL) experiments (Figure 9C–F). A low reduction in the number of identified peptides in these two experiments when using FAIMS was also observed due to the increased time of each mass spectrometry cycle and the CV filtering. Importantly, the decrease in the number of peptides resulted in a limited increase—up to 26%—in the number of protein identifications and quantifications, which was also accompanied by a decrease in protein coverage, as previously observed for the cell extract proteomics experiment (Figure 9G–J).

These results showed that the proteomics improvements observed here with the FAIMS technology for the analysis of paired FFPE and exosomes isolated from low plasma volumes (250 µL) were related to the nature and quality of the biological samples. Thus, our results highlight and confirm the relevance of FAIMS for the proteomics analysis of deteriorated, fragmented, and/or cross-linked protein samples, or those where the material for isolation was scarce and the final material contains highly abundant proteins that reduce the dynamic range of the protein samples, such as FFPE tissues or exosomes isolated from low plasma volumes, respectively. Therefore, our results highlight that these samples require the additional level of separation (ion filtering) provided by FAIMS, whereas it shows limited benefits for non-cross-linked and non-deteriorated biological samples.

## 4. Discussion

TMT is one of the most widely used quantitative proteomics techniques for the identification of dysregulated proteins, as the analysis of multiple samples in a single experiment increases the accuracy of the assays, avoiding bias due to sample preparation and processing while reducing the time and costs associated with the proteomics experiments [45,46,47]. An in-depth characterization of the proteome associated with a biological sample would increase the number of altered proteins identified in a TMT experiment, which might be of high interest for further analysis of the pathology of the study or as biomarkers or therapeutic targets of intervention. However, the complexity of the different biological samples that can be analyzed by TMT mass spectrometry (e.g., plasma, tissue, or cell protein extract samples) might hinder their processing, even when using state-of-the-art mass spectrometers, thus affecting the accuracy and sensitivity of the TMT analyses and reducing the efficiency of the experiment.

In 2006, high-field asymmetric waveform ion mobility spectrometry technology (FAIMS) appeared to surpass the detection limits of LC-MS/MS analyses when coupled to mass spectrometers by improving the in-depth knowledge of the proteome and the characterization of proteoforms [48,49,50,51]. FAIMS technology allows for the isolation of the ions of interest and the reduction of background ions by applying a CV or CVs of interest, thus reducing protein complexity. Thus, in bottom-up proteomics, FAIMS was suggested to improve the proteome and proteoform coverage of complex protein mixtures by increasing peptide separation, reducing peptide co-isolation, and filtering the ions of interest [18,52,53]. Moreover, the increased depth into the proteome obtained with FAIMS was associated with reduced peptide identification and protein coverage, as many peptide ions are ejected at the entrance of the mass spectrometer. Furthermore, FAIMS has been reported to improve the sensitivity of limited samples in combination with monolithic capillary columns using either an Orbitrap Fusion Lumos or disposable trap columns, Evosep One, and Orbitrap Exploris 480 from 10, 100, and 1000 processed cells, or approximately 600 ng of peptide digested from microdissected FFPE [54,55]. Additionally, FAIMS on an Orbitrap Eclipse Tribrid mass spectrometer has also been shown to be useful for single-cell proteomics [56] and targeted proteomics workflows [57]. Despite these reported improvements in increasing the depth into the proteome, proteoforms, and for the identification of PTMs [58], imaging mass spectrometry [59], or PRM analyses [57,60], this technology has not been widely accepted among the proteomics community because of the cost, the time consumed, and the belief that peptide loss is accompanied by reduced protein identifications. Importantly, most of the works investigating the benefits of FAIMS took advantage of the analysis of HeLa commercial peptide extracts to set up the mass spectrometers’ conditions and then analyzed the sample of interest, comparing it with the equipment without FAIMS [18,55,61,62]. Even more, to date, almost no reports compare the performance of the mass spectrometers with or without FAIMS for the analysis of clinical samples.

Here, we have tried to shed some light on the benefits of FAIMS for the analysis of clinical samples, comparing the performance of the Orbitrap Exploris 480 with and without FAIMS. To this end, we have investigated the benefits of FAIMS for the TMT proteomics analysis of delicate, cross-linked, and fragmented FFPE tissues and plasma exosomes from CRC patients in comparison to more robust cell protein extracts. Quantification in the Orbitrap Exploris 480 mass spectrometer was also optimized for TMT-based proteomics analyses enabled by the fast-scanning turboTMT method. Plasma exosomes are valuable samples in the study of a disease as they are enriched in proteins involved in cell–cell communication and thus might have an important role in the development of a pathology [63,64,65,66]. However, the MS analysis of these samples is of high complexity due to the low abundance of the proteins contained in the exosomes and because these samples can include other highly abundant plasma proteins, which might mask the identification of exosome proteins. This would reduce the dynamic range of the MS analyses and the number of peptide and protein identifications and quantifications. In addition, MS analyses of exosomes are also limited by the reduced amount of sample that can be obtained. Regarding tissues, these paired samples provide important information about the development and progression of a pathology, as they encompass not only pathological cells but cells and other molecules from the microenvironment, which in most cases have an important role in a pathology (e.g., the tumoral microenvironment in cancer) [67,68]. In contrast to frozen or fresh tissue samples, which are easy to analyze by proteomics and other orthogonal techniques (i.e., western blot analysis), proteins from formalin-fixed paraffin-embedded tissues suffered from extensive protein cross-linking, which reduces the sensitivity of MS analyses and avoids their analysis by western blot. However, FFPE tissues are commonly used in clinical diagnostic routines as they allow for the long-term storage and complete preservation of tissues [2,3,4]. Due to the high value of these samples, the optimization of mass spectrometry methods for an in-depth proteomics characterization of paired FFPE tissues and plasma exosomes isolated from scarce samples is of high interest. In contrast to exosomes and FFPE tissues, cell protein extracts are easy to analyze by mass spectrometry, as large amounts of fresh cell protein extracts can be easily obtained and because proteins extracted from cells or fresh frozen tissue do not usually present modifications that might disrupt peptide identifications as FFPE cross-linked or fragmented proteins.

Here, we observed that mass spectrometry analyses without FAIMS of paired FFPE tissues or exosomes isolated from 250 µL of plasma resulted in poor coverage of the proteome (less than 1000 and 300 proteins identified and quantified, respectively). However, identified and quantified proteins increased significantly in our hands when FAIMS was coupled to the Orbitrap Exploris 480 mass spectrometer (>100% increase in protein identifications and quantifications). In contrast, the protein extracts from cells, brain tissue, or exosomes isolated from large plasma volumes (3 mL) analyzed with or without FAIMS resulted in about 5000, 3500, and 1500 identified and quantified proteins, respectively. Importantly, as a similar amount of protein extracts from the clinical samples (FFPE tissue and exosomes) or cells was used (Appendix A), the differences observed in the number of identified peptides and proteins should not be related to the amount of protein extracts used for the TMT analyses but to the nature, fragmentation, and/or cross-linking of the protein samples. In addition, approximately 800 ng were injected into each fraction in the Orbitrap Exploris 480, getting intensity signals ranging from 10^9^ to 10^10^, thus indicating that the differences observed here with and without FAIMS technology could not have been surpassed by injecting higher peptide amounts into each fraction. Additionally, the protein content of exosomes contained three dominant bands around 70 to 100 kDa that could also mask the detection of more proteins, thus reducing the dynamic range of the proteomics analysis of exosome samples. Accordingly, the significant increase in the number of identified peptides and proteins in the paired clinical samples with FAIMS was related to the reduced co-isolation of precursor ions (peptides), which is supported by the extra separation and filtering of peptides at the entrance of the mass spectrometer due to the selected CVs. Thus, FAIMS technology increased the accuracy and dynamic range of complex, cross-linked, and/or scarce biological samples. In contrast, MS analyses of protein extracts from cells, frozen tissue, and exosomes isolated from 3 mL of plasma without FAIMS resulted in a high proteome coverage, which was slightly increased (10–26% increment) with FAIMS. In addition, a decrease in the number of peptide identifications in the analysis of these latter TMT experiments with FAIMS was observed in comparison with the analysis without FAIMS. This might be related to the increase in the time of each mass spectrometry cycle when using more than one CV and to the selection of subsets of precursor ions of interest. Contrary to cell and tissue samples, the number of identified peptides significantly increased for the FFPE tissues and exosomes from 250 µL of plasma experiments, highlighting the potential of FAIMS technology to increase the proteome coverage of FFPE tissues and/or exosomes isolated from scarce clinical samples. These results demonstrate that the extra peptide filtering associated with FAIMS reduces the number of precursor ions that reach the mass spectrometer but allows for an increased peptide separation that is mandatory in these “hard-to-analyze” samples. Thus, in these samples, the peptide losses associated with increased mass spectrometry cycles and CV filtering with FAIMS are highly compensated by increased peptide fractionation, which significantly reduced the number of overlapping and masking peptides. In addition, the benefits of FAIMS to increase the proteome coverage of actual paired FFPE and exosome clinical samples were reflected in the higher number of identified dysregulated proteins in each TMT experiment with FAIMS. In addition, the high correlation observed between paired tissue and plasma exosomes from patients suggested that some proteins associated with the disease identified in tissue samples might also be measured in plasma exosomes, thus allowing their detection by minimally invasive techniques in patients. Although robust mass spectrometry data can be obtained without FAIMS for easy-to-work samples, such as protein extracts from cancer cells, frozen tissue, or exosomes isolated from large plasma volumes, the use of FAIMS with these samples is not contraindicated. In this sense, we observed a limited improvement with FAIMS in the number of protein identifications and quantifications (up to 26%) in these samples. There, the higher separation of peptides achieved by FAIMS might significantly reduce peptide co-elution, thus increasing the number of peptide identifications when the starting material is scarce. Remarkably, according to our results, the use of FAIMS would be highly recommended with deteriorated and/or cross-linked protein samples, with samples with a high ratio of highly abundant contaminant proteins, or with samples derived from very low starting materials, such as paired FFPE and exosome clinical samples.

One of the interesting results of the study was also related to the use of two CVs or three CVs for the proteomics analyses of clinical samples. Importantly, no significant differences were observed in the number of identified and quantified peptides and proteins with three CVs regarding the hard-to-analyze TMT samples. Since the use of three CVs requires a three-hour gradient for the analysis of each TMT experiment, the slight differences observed in the proteome coverage with three CVs in comparison with two CVs do not compensate for the increased time consumed and reagent expenses required per experiment. Thus, the FAIMS Pro Duo Interface with two CVs should be recommended for the analysis of both easy-to-work and hard-to-analyze TMT samples, as it allows for increased proteome coverage without increasing the time of the assay, which might aid in the identification of dysregulated proteins associated with a pathology.

Finally, as potential limitations of the study, this work has only been performed with the Orbitrap Exploris 480, and thus, other analyses using other mass spectrometers such as the Orbitrap Exploris 240, Eclipse, or Fusion Tribrid mass spectrometers should also be performed. This would allow us to compare the data obtained with the Orbitrap Exploris 480 and determine whether the benefits of the FAIMS Pro Duo Interface would also be extensive to other mass spectrometers for the analysis of deteriorated, cross-linked, scarce, or damaged clinical and biological samples.

## 5. Conclusions

Here, we demonstrated that the application of FAIMS with two CVs on an Orbitrap Exploris 480 mass spectrometer in data-dependent acquisition-based shotgun proteomics experiments increased the proteome coverage. When analyzing TMT-based proteomics analyses of FFPE tissue and exosomes isolated from low plasma volumes (250 µL), the protein identification and quantification increases were 205% and 130%, respectively. Additionally, when analyzing TMT-based cell protein extracts, frozen tissue protein extracts, or protein extracts from exosomes isolated from large plasma volumes (3 mL), proteome coverage was maintained, with a limited increase ranging from 10 to 26% in the number of proteins identified and quantified. Therefore, FAIMS might be considered for its routine use to increase the coverage of the proteome and proteoforms with cross-linked and deteriorated protein extracts, such as those derived from FFPE tissue samples. Additionally, FAIMS should also be considered when the material is scarce and the dynamic range of the mass spectrometry analysis is expected to be low due to the presence of highly abundant proteins, such as the exosomes isolated here from low plasma volumes.

Importantly, FAIMS enabled the analysis of minimal sample amounts by effectively removing interfering single-charge background ions, thus improving the signal-to-noise ratio. Here, we demonstrate that it is possible to quantify more than 2000 proteins from deteriorated or cross-linked FFPE-derived tissue protein extracts, highlighting its potential for applications such as laser microdissected clinical samples. Moreover, we also envisioned that the use of three CVs instead of two CVs would also produce an increase in proteome coverage in complex samples. However, we observed an insignificant increase in the coverage of the proteins with the three CVs in comparison to two CVs, thus supporting the use of just two CVs to reduce the time of the assay, the cost of reagents, and the time consumed by the mass spectrometer per run. Consequently, as future research directions, we envision that FAIMS would find widespread use to increase the coverage of the proteome and proteoforms in proteomics laboratories specifically for the analysis of protein extracts derived from deteriorated, cross-linked, or fragmented clinical specimens, or from protein extracts derived from really scarce material, PRM and PTM analyses [57,58,60], or samples for single cell proteomics analyses.

## Figures and Tables

**Figure 1 proteomes-11-00035-f001:**
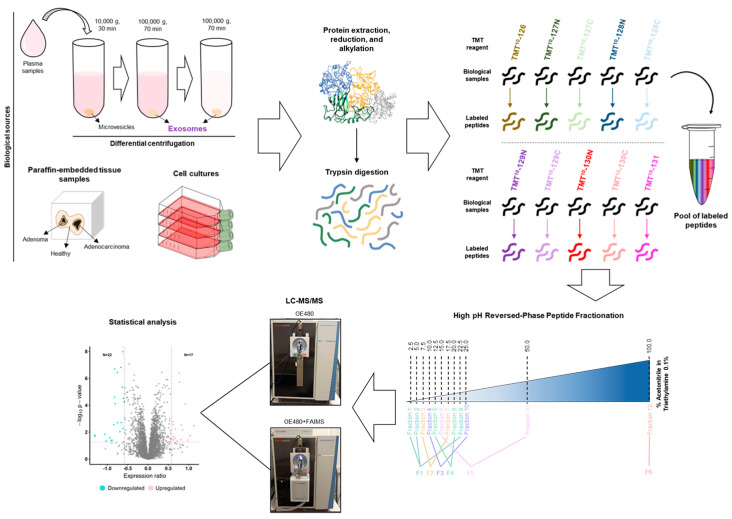
Workflow for the mass spectrometry analysis of paired plasma exosomes, FFPE tissue samples, and cultured cell protein extracts in an Orbitrap Exploris 480 (OE480) equipped or not with FAIMS Pro Duo Interface. After protein extraction, denaturation, reduction, alkylation, and trypsin digestion, peptides were TMT labeled and separated according to their hydrophobicity using high-pH reverse-phase columns and an ACN gradient. Differential peptide fractions were pooled together into six fractions, which were subsequently analyzed in the OE480 with or without FAIMS Pro Duo Interface to determine the benefits of FAIMS for the analysis of exosomes and FFPE samples. Finally, statistical analysis was performed to identify dysregulated proteins associated with CRC. Grey dots represent proteins that do not fulfill the criteria for being classified as significantly dysregulated proteins.

**Figure 2 proteomes-11-00035-f002:**
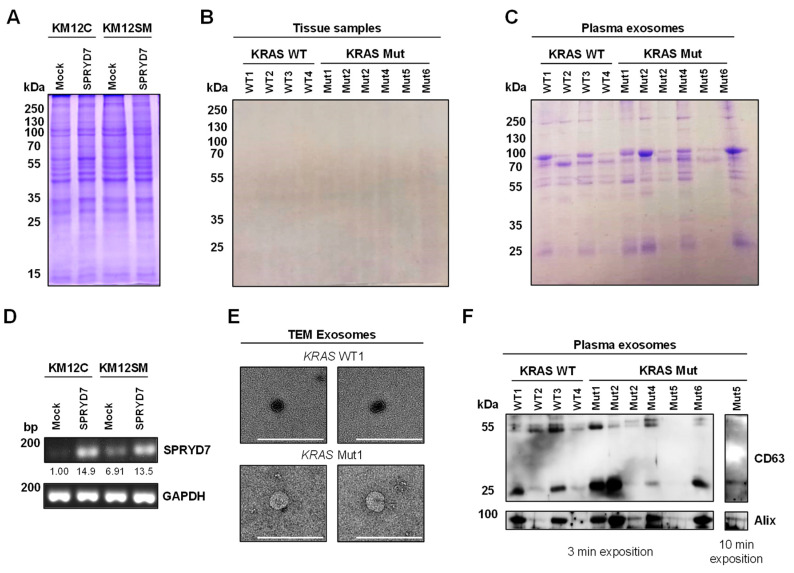
Quality control of the protein extracts used for proteomics. Coomassie blue staining of cell extracts (**A**), FFPE tissue samples (**B**), and plasma exosomes (**C**) confirmed the quality of the samples for the proteomics analyses. (**D**) The stable overexpression of SPRYD7 in KM12C and KM12SM CRC cells was confirmed by PCR in comparison to mock-stable transfected cells (control). (**E**) Transmission electron microscopy (TEM) of plasma extracellular vesicles isolated by differential centrifugation revealed vesicles smaller than 150 nm compatible with exosomes. Scale bar: 200 nm. (**F**) Western blot analysis of plasma exosomes showed the presence of Alix and CD63, two proteins specific to extracellular vesicles, confirming the quality of the exosome samples. Two different exposition times are depicted to visualize CD63 and Alix in all samples. WT: wild-type *KRAS*; Mut: mutant *KRAS*.

**Figure 3 proteomes-11-00035-f003:**
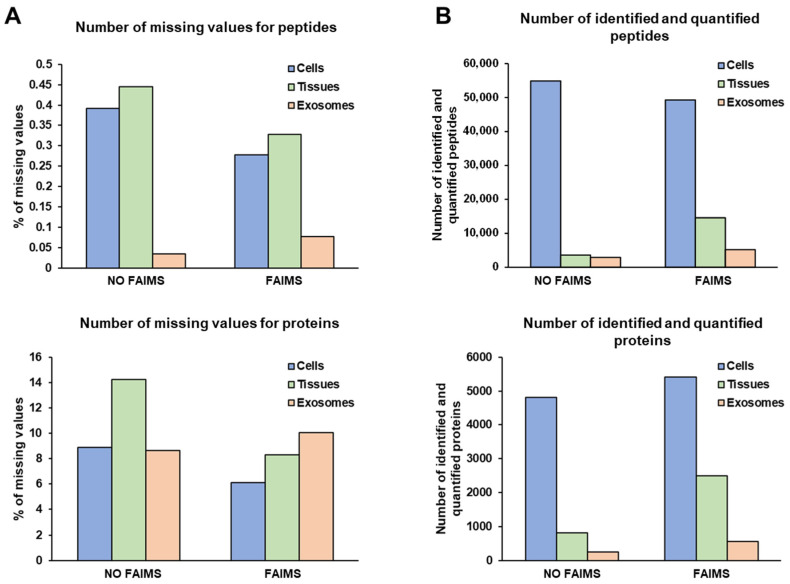
Peptide and protein identifications and quantifications. (**A**) The number of peptide (**top**) and protein (**bottom**) missing values from the three TMT experiments with or without FAIMS Pro Duo Interface was lower than 1% and 15%, respectively, highlighting the efficiency of the TMT labeling. (**B**) A significant increase (>70%) in the number of identified and quantified peptides (**top**) and proteins (**bottom**) was obtained for the FFPE tissue and plasma exosomes TMT experiments with FAIMS Pro Duo Interface technology, whereas a 10% increase in the number of proteins was obtained for the cell TMT experiment with a slight reduction in the number of identified peptides.

**Figure 4 proteomes-11-00035-f004:**
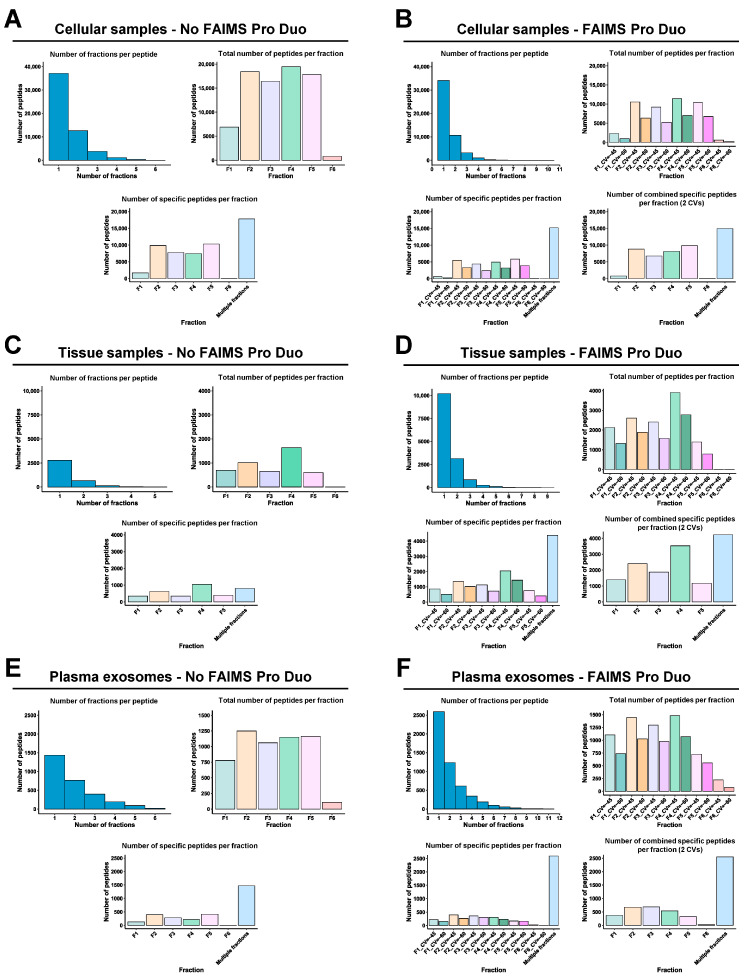
Peptide analysis of each TMT experiment. The number of fractions where each peptide was identified, the number of total peptides per fraction, and the number of specific peptides per fraction without (**A**,**C**,**E**) or with (**B**,**D**,**F**) FAIMS Pro Duo Interface highlighted the increased peptide separation obtained with FAIMS.

**Figure 5 proteomes-11-00035-f005:**
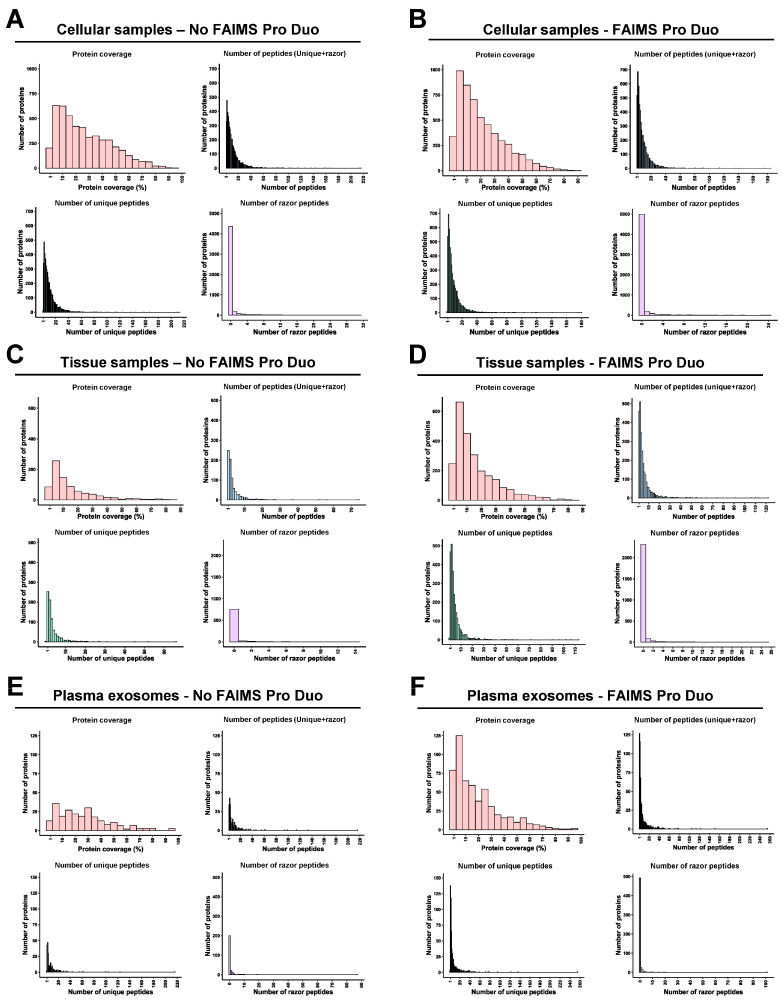
Protein analysis of each TMT experiment. No differences were obtained in the protein coverage and number of total, unique, and razor peptides per protein for the cell extract experiment without (**A**) or with (**B**) FAIMS. However, the protein coverage and the number of total, unique, and razor peptides per protein decreased for the plasma exosome and FFPE tissue samples without FAIMS (**C**,**E**) in comparison with their mass spectrometry analysis with FAIMS (**D**,**F**).

**Figure 6 proteomes-11-00035-f006:**
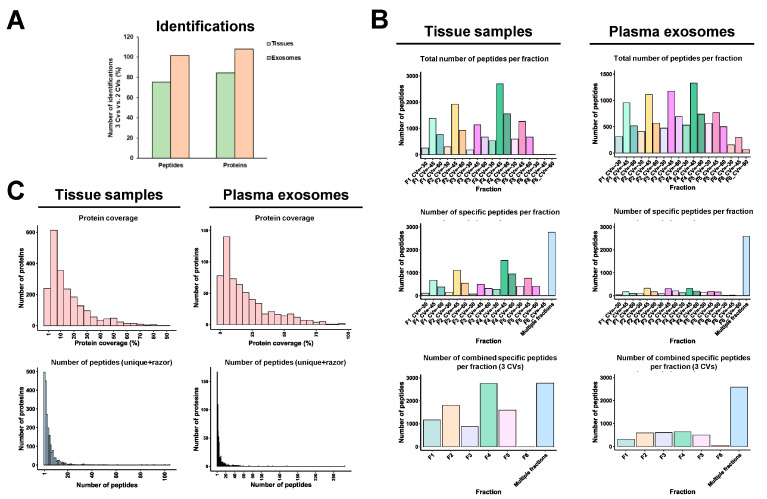
FAIMS analysis with three CVs of hard-to-analyze samples. (**A**) Non-significant improvements in the number of identified and quantified peptides and proteins were obtained with three CVs in comparison with two CVs. (**B**) Slight differences in the total number and specific number of peptides per fraction were obtained with three CVs in comparison with two CVs for the FFPE tissues (**left**) and plasma exosomes (**right**) samples. (**C**) Similar protein coverage and number of peptides per protein were obtained with the three CVs in comparison with the two CVs.

**Figure 7 proteomes-11-00035-f007:**
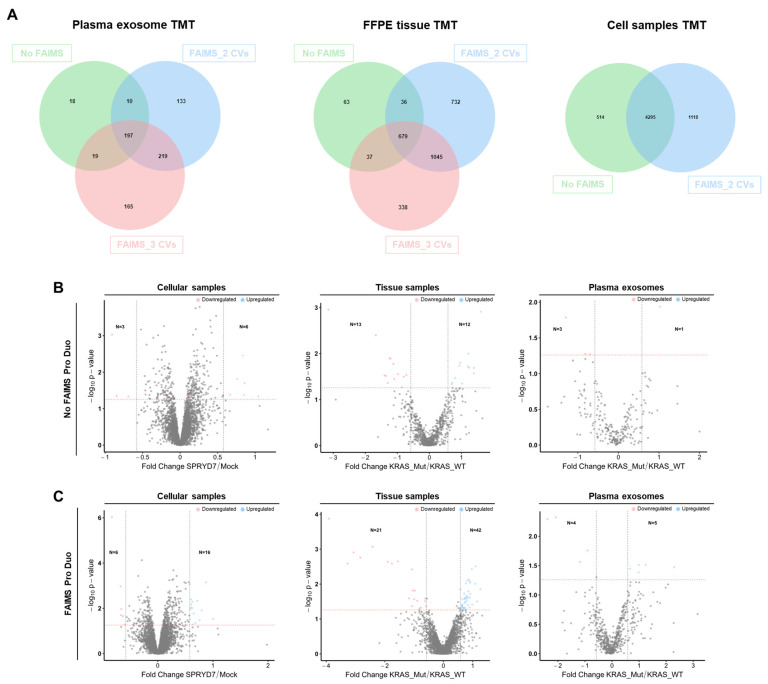
Statistical analysis of mass spectrometry data. (**A**) The Venn diagram of the indicated TMT experiments shows the analysis of the number of identified proteins with or without FAIMS and comparing two and three CVs. Statistical analysis for the identification of dysregulated proteins in the indicated TMT experiments, comparing the results without (**B**) or with (**C**) FAIMS Pro Duo Interface. The increase in the number of protein identifications was reflected in an increased number of dysregulated proteins in the three TMT analyses. The x-axis represents the log_2_ expression ratio (fold change) of protein expression differences between the two groups of study. The y-axis represents the −log_10_
*p* value. Colored dots represent differentially expressed proteins upregulated (blue, expression ratio ≥ 1.5) and downregulated (red, expression ratio ≤ 0.58) in CRC cells overexpressing SPRYD7 or in CRC patients with mutated *KRAS* with a *p*-value < 0.05. *p*-value = 0.05 is represented by a red dashed horizontal line, whereas the 1.5-fold expression difference is represented by two black dashed vertical lines. Grey dots represent proteins that do not fulfill the criteria for being classified as significantly dysregulated proteins.

**Figure 8 proteomes-11-00035-f008:**
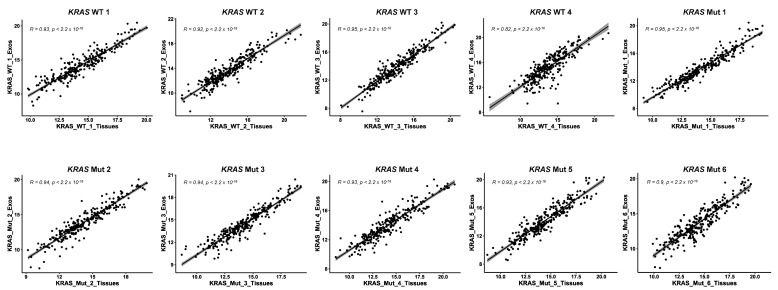
Correlation analysis between paired FFPE and exosome samples. Individual Pearson correlation analysis among paired individual FFPE and exosome samples from the 10 patients used in the study showed a high correlation in the protein levels in both samples, highlighting that dysregulated proteins could become CRC blood-based biomarkers. WT: wild-type *KRAS*; Mut: mutant *KRAS*. R: Pearson correlation coefficient; *p*: *p*-value.

**Figure 9 proteomes-11-00035-f009:**
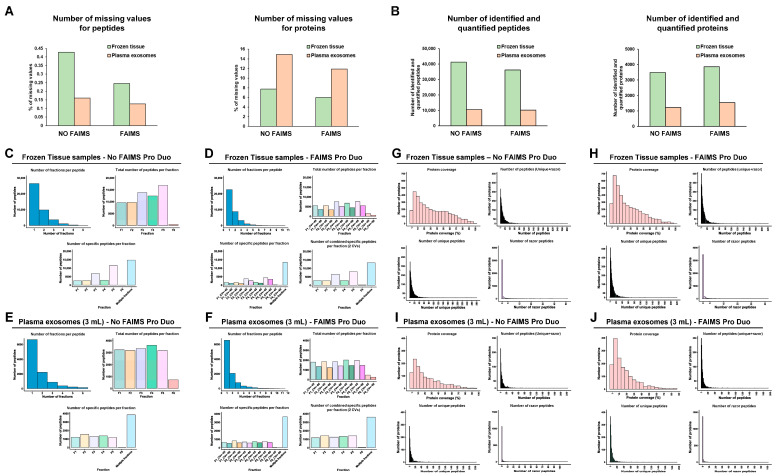
MS data analysis of the TMT experiments with non-cross-linked and/or non-deteriorated protein samples from frozen tissue and plasma exosomes isolated from large plasma samples. (**A**) A low number of missing values was obtained in both TMT experiments for the identified peptides (**left**) and proteins (**right**). (**B**) A 12% reduction and 11% increase in the number of identified and quantified peptides (**left**) and proteins (**right**), respectively, were obtained for the brain tissue protein extracts, whereas these numbers were 3% and 26%, respectively, for the exosomes isolated from large plasma volumes (3 mL) experiment. The analysis of the number of fractions where each peptide was identified, the number of total peptides per fraction, and the number of specific peptides per fraction without (**C**,**E**) or with (**D**,**F**) FAIMS Pro Duo Interface highlighted the increased peptide separation obtained with FAIMS technology. The analysis of the proteins identified and quantified in each TMT experiment revealed no changes in the number of total, unique, and razor peptides obtained without (**G**,**I**) or with (**H**,**J**) FAIMS technology, whereas a slight reduction in the protein coverage was observed in both experiments.

## Data Availability

The data presented in this study are available in the main body of the manuscript and in the Appendix A. Proteomics data are deposited at the ProteomeXchange Consortium repository via PRIDE under the datasets PXD042596, PXD042601, PXD045367, PXD045362, and PXD042597 [69].

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
