# Peer review of "Benefits of FAIMS to Improve the Proteome Coverage of Deteriorated and/or Cross-Linked TMT 10-Plex FFPE Tissue and Plasma-Derived Exosomes Samples"

_proteomes, 2023, doi:10.3390/proteomes11040035_

Round 1

Reviewer 1 Report (Previous Reviewer 1)

Dear Authors, 

I am writing to share my comments regarding your manuscript entitled, "Benefits of FAIMS to improve the proteome coverage of delicate, deteriorated, and/or cross-linked TMT 10-plex FFPE tissue and plasma-derived exosomes samples."

Having thoroughly reviewed the revisions you made in response to the feedback provided, I am delighted to express my satisfaction with the improvements. The changes made, and the extra work put into the manuscript have significantly enhanced its quality, readability, and impact. You have shown an excellent commitment to the rigorous scientific process. The effort you put into addressing every concern and suggestion demonstrates a high level of integrity in your research practices. The manner in which you approached the feedback is not just commendable but also exemplary.

The study, as it stands now, presents a comprehensive, sound methodology, and clear, coherent results. Your diligence in making the requested revisions has only amplified the value and relevancy of your study, thereby increasing its potential impact on the field.

I am confident that the insights provided in your research, particularly with regard to the benefits of using FAIMS in TMT proteomics of FFPE tissues and plasma exosomes, will be well received by the scientific community. This manuscript is now a strong contribution to the field, and I commend you for your hard work. Once again, congratulations on a job well done. I look forward to seeing this manuscript published and anticipate that it will generate meaningful dialogue in our shared field of proteomics. 

Best

Author Response

We are very grateful to the reviewer for his/her work, and for his/her comments during the peer review process. 

Reviewer 2 Report (Previous Reviewer 2)

The revised manuscript has improved significantly in scientific and grammar presentation. The improved results section communicated clearly the rational, the findings and conclusions of the study. I recommend this manuscript to be published. 

Author Response

We are very grateful to the reviewer for his/her work, and for his/her comments during the peer review process. 

Reviewer 3 Report (Previous Reviewer 3)

I reviewed the original version of this manuscript, and now the revised copy.  To put things bluntly, it is clear that the authors did not appreciate my concerns.  While they provide revisions, noting in particular the qualification of "delicate, deteriorated, and/or cross-linked samples" no actual additional work has been done.  My concern remains that the results are 'interesting' but that there is no scientific basis to support them. I am not interested in speculative assumptions.  All of my suggestions were to allow the authors to realize that they could have performed extremely simple experiments to test these hypotheses and try to explain their anomalous results.  Without this proof, the paper is nothing more than a summary of unusual findings. 

I cannot recommend publication of this manuscript. It is incomplete, and would serve no real value to the proteomics /mass spectrometry community until further work is completed.

Author Response

Reviewer #3

Comments and Suggestions for Authors

Comments on August 1st, 2023

I reviewed the original version of this manuscript, and now the revised copy. To put things bluntly, it is clear that the authors did not appreciate my concerns.  While they provide revisions, noting in particular the qualification of "delicate, deteriorated, and/or cross-linked samples" no actual additional work has been done.  My concern remains that the results are 'interesting' but that there is no scientific basis to support them. I am not interested in speculative assumptions. All of my suggestions were to allow the authors to realize that they could have performed extremely simple experiments to test these hypotheses and try to explain their anomalous results.  Without this proof, the paper is nothing more than a summary of unusual findings. 

I cannot recommend publication of this manuscript. It is incomplete, and would serve no real value to the proteomics /mass spectrometry community until further work is completed.

We are very grateful to the reviewer for his/her work, and for his/her comments during the peer review process. Indeed, to avoid any misunderstanding we asked for further comments to better understand the first and second round of revision. We think that below we have now addressed the comments of the reviewer.

We hope that with the inclusion of the new experiments we will be able to satisfy the concerns of the reviewer. Furthermore, regarding the qualification of “delicate, deteriorated and /or cross-linked samples” we have removed from the title “delicate” since we agree with the reviewer that most, if not all, proteomics samples could be considered as delicate. Moreover, in this sense we have also carefully edited the revised version of the manuscript to avoid any misunderstanding regarding deteriorated and/or cross-lined protein samples.

Comments on August 29th, 2023

First, we would like to thank the reviewer for helping us to clarify their concerns and suggestions for the manuscript by providing us with more detailed comments, which we have addressed as indicated below.

In my initial review, I mentioned that the results were certainly interesting, but that the study was incomplete. Generally speaking, the authors described an experiment that provided improved results under "certain cases", but had the opposite effect for other cases. They provided no explanation as to why that was. In the revised manuscript, the authors have now qualifying their results according to difference between sample types. Hence, the title now adds "delicate, deteriorated, and/or cross linked" to qualify cases where they saw improvements. This is still not a sufficient conclusion, as it has yet to provide evidence to explain their results. Just to pick an example what exactly qualifies as a "delicate" proteomic sample? Could it not be stated that all biological samples are delicate? This is a rather subjective term. To follow that, what qualifies as "deteriorated"? Placing the term "or" in the title implies that any "deteriorated" sample would benefit from FAIMS.

In the revised version of the manuscript, we have clarified those cases where FAIMS shows improvements. To this end, we have compared the performance of the deteriorated and/or cross-linked protein extracts with that of frozen tissue and exosomes isolated from large plasma volume. Therefore, we now provide evidence that FAIMS provides better results with protein extracts from deteriorated samples, cross-linked samples, or those from exosomes isolated from scarce plasma volumes where the highly abundant proteins are present in a higher ratio in comparison to exosome proteins, in contrast to the cell protein extracts, fresh frozen tissue protein extracts, or exosomes isolated from large plasma volumes.

Moreover, we agree with the reviewer that all protein extracts used for proteomics could be considered as delicate, and thus we have removed this from the title and carefully edited the manuscript to avoid this term. Furthermore, we have indicated in the revised version of the manuscript that FAIMS shows specific benefits for the proteomics analysis of FFPE tissues and/or plasma samples isolated from low plasma volumes (250 µL).

My initial review provided specific examples of experiments that could have been performed, to test a hypothesis. But I will add that those experiments were only meant as suggestions, and not meant to inform the authors as to which experiments were essential. The appropriate experiment depends entirely on the hypothesis. For example, if it relates to sample loading only, the obvious experiment is to vary the amount of sample. If it's because of non-protein contaminants, either remove them, or spike them into a control. Cross linking? Start with a fresh sample... then cross link it. Sample complexity? Prefractionate... I simply point out that unusual results require some form of explanation. And that explanation cannot simply amount to speculation.

To address the concern of the reviewer, we have included in the revised version of the manuscript two new TMT experiments to compare the performance of deteriorated and/or cross-linked protein extracts with that of fresh frozen tissue protein extracts and exosomes isolated from large plasma volumes to compare cross-linked versus non-cross-linked protein extracts and exosomes isolated from paired clinical samples from low volume plasma samples and from large plasma volume.

In these experiments performed with the same amount of protein extracts than the previous TMTs depicted in the original manuscript (10 µg of each protein extract), we observed that the analysis in the Orbitrap Exploris 480 with or without the FAIMS of frozen non-cross-linked protein tissue extracts maintained the number of peptides and proteins identified and quantified at a similar extent. The same performance was almost observed with exosomes isolated from large plasma volumes (3 mL). In both cases, a 12% and 3% reduction in the number of identified and quantified peptides was observed from frozen tissue protein extracts and exosomes isolated from large plasma volumes, and an 11% and 26% increase in the number of identified and quantified proteins, in concordance with the results obtained for the cell extracts in the original version of the manuscript. These results were considerably lower to that observed comparing with the increase of 205% and 130% of identified and quantified proteins from cross-linked FFPE derived protein extracts and paired exosomes isolated from low plasma volumes, respectively. Therefore, according to these results FAIMS is highly recommended for the analysis of cross-linked and deteriorated samples or exosomes isolated from low plasma volumes (lower or equal to 250 µL) containing highly abundant proteins masking the analysis by proteomics. In addition, we have also highlighted in the revised version of the manuscript that a higher dynamic range in the MS experiments is achieved with the FAIMS technology, which has been here demonstrated in the two TMT experiments with plasma exosomes isolated from large and low plasma volumes in which most plasma contaminant proteins were present (see Coomassie blue staining from Figure 2C and Figure S3C).

Finally, we also want to highlight that TMT experiments were performed in all cases with 10 µg of protein extracts and we loaded into the Orbitrap Exploris 480 approximately 800 ng of peptide samples per analyzed fraction, and thus, we do believe hat we would not get improved results either regarding sample loading because we cannot saturate the MS analysis (signal intensities in all cases ranged between 109-1010) or varying the amount of samples in the TMT experiments because we are working with the recommended concentration of protein in these experiments according to the manufacturer.

In this context, and to address the concern of the reviewer, we have included all new information in the revised version of the manuscript highlighting in the abstract, results, and discussion that FAIMS is recommended for the proteomics analysis of deteriorated and/or cross-linked protein samples. Additionally, we have also included in the revised version of the manuscript a new figure in the main body of the manuscript (Figure 9), and a new supplementary figure (Figure S3). Moreover, since we observed a limited increase in the performance of the MS analysis of fresh frozen tissue protein extracts, cell protein extracts, and exosomes derived from large plasma with FAIMS, we have also highlighted that the benefits of FAIMS are not limited to the analysis of deteriorated or cross-linked protein extracts but that for this kind of samples is highly recommended. However, for those protein extracts non-deteriorated or non-cross-linked the benefits of FAIMS should not justify their use because of the expenses related to time-consume in the mass spectrometer, the cost of the reagents, and the reduced protein coverage achieved due to the reduction in the number of identified peptides.

As it stands, the pairing of FFPE tissues and exosomal material may make it difficult (at least, in my opinion) to arrive at an explanation that justifies their findings. These are very different samples. Perhaps there is a connection between the two. Perhaps it is just coincidental. Perhaps the authors should focus on one.

We do agree that both samples are actually different in nature and in protein content, and to get to an explanation that justifies both findings might be difficult. However, we do believe that the inclusion of these kind of samples in the same manuscript should help the readers to see that FAIMS could help not only for FFPE derived samples but also in those protein extracts obtained from really scarce samples, as exosomes isolated from 250 µL plasma samples, while containing at the same time abundant proteins, which with the analysis without FAIMS would be difficult to get any result. In addition, both FFPE tissues and low plasma volumes are commonly provided by hospitals for scientific research, and thus we further believe that the inclusion of both type of samples in this work would help proteomics researchers and research community to improve their sample processing and data analysis workflows.

Therefore, regarding this comment we have maintained in the revised version of the manuscript the analysis of both kind of samples. However, to address the concern of the reviewer we have highlighted in the revised version of the manuscript that FAIMS is useful not only for cross-linked FFPE tissue protein extracts (this work) and the analysis of protein extracts from exosomes isolated from low plasma volumes (this work) containing highly abundant proteins masking the analysis without FAIMS, but also for imaging mass spectrometry [1], PRM analysis [2], or for the analysis of PTMs [3].

Finally, I will add that there may already be some insight from the literature to help explain these results. If that is the case, then additional experiments may not be required. Then again, if the answer is already published, the authors must make it clear what is novel about their current study.

Thank you for the comment. To address the concern of the reviewer, additionally to the experiments performed, we have carefully revised here the current literature with FAIMS to include also relevant manuscripts above indicated to help addressing the concerns of the reviewer.

I hope this is a satisfactory justification for my position on this manuscript.

Finally, we want to thank again the reviewer for his/her inputs and for the additional effort to clarify us his/her position on the manuscript. We should also highlight that we really appreciate the comments of the reviewer along the whole peer review process, which we believe we have now addressed in the revised version of the manuscript. These comments have significantly strengthened the revised version of the manuscript.

References

  1. Griffiths, R.L.; Simmonds, A.L.; Swales, J.G.; Goodwin, R.J.A.; Cooper, H.J. LESA MS Imaging of Heat-Preserved and Frozen Tissue: Benefits of Multistep Static FAIMS. Anal Chem 2018, 90, 13306-13314, doi:10.1021/acs.analchem.8b02739.
  2. Sweet, S.; Chain, D.; Yu, W.; Martin, P.; Rebelatto, M.; Chambers, A.; Cecchi, F.; Kim, Y.J. The addition of FAIMS increases targeted proteomics sensitivity from FFPE tumor biopsies. Sci Rep 2022, 12, 13876, doi:10.1038/s41598-022-16358-1.
  3. Adoni, K.R.; Cunningham, D.L.; Heath, J.K.; Leney, A.C. FAIMS Enhances the Detection of PTM Crosstalk Sites. J Proteome Res 2022, 21, 930-939, doi:10.1021/acs.jproteome.1c00721.

Reviewer 4 Report (Previous Reviewer 4)

The resubmitted version of the manuscript shows significant improvement. The authors have addressed additional aspects, rendering the manuscript sound and complete. Particularly, the inclusion of more statistical analysis methods enhances the clarity of how the data was collected. Moreover, the authors have provided a more profound and explicit discussion regarding the advantages of utilizing FAIMS for challenging samples such as tissue or exosomes, which are characterized by higher crosslinking and detection difficulties. Furthermore, the manuscript now includes a thorough comparison of 2CV and 3CV, highlighting that there is no significant difference between the two, but also indicates that 3CV is unnecessary, considering the additional time and reagent costs it incurs.

Overall, the improvements made in this resubmission have significantly strengthened the manuscript and will contribute to its scientific value.

Line 311, at the end of the sentence, should remove the full stop. 

Author Response

We have fixed the typo indicated by the reviewer.

We are very grateful to the reviewer for his/her work, and for his/her comments during the peer review process. 

This manuscript is a resubmission of an earlier submission. The following is a list of the peer review reports and author responses from that submission.

Round 1

Reviewer 2 Report

In this manuscript, the authors demonstrated the use of FAIMS technology in conjunction with LC/MS for a more sensitive peptide detection in FFPE and patient-derived exosome samples. The rational of the paper was well designed and the experimental results were scientifically sound and presented clearly. I only have minor suggestions below.

Introduction: The authors could highlight the technical advantages of FAIMS a bit more, esp. in the context of dealing with delicate, damaged and small quantities of clinical samples.

Fig 2C: The staining pattern between different samples are quite different. Can authors comment on the dominant bands around 70-100kD with regard to their relevance to KRAS or CRC biology?

Fig 2F: It looks like KRAS mut sample 4 doesn't have CD63 or Alix expression. Can authors comment on the quality of that sample and the rationale to include it in the analysis?

Fig 4 and 5: I'd recommend to have No FAIMS and FAIMS on the same plot for direct, visual comparison. It's not easy to appreciate the superior performance of FAIMS from the current figure configuration.

Reviewer 3 Report

The authors  examine the question whether FAIMS can improve proteome analysis of 'difficult to work with' samples, and explore FFPE and exosomal materials, derived from clinically relevant samples.  Their main finding is that FAIMS significantly improves the number of quantified peptides and proteins for these two samples, though not for 'easy to work with' samples that have already yielded strong proteomics data in the absence of FAIMS.

I find this to be an interesting, though extremely peculiar result.  I don't disagree with the findings. However, I am not satisfied with the study as a whole, as there lacks a justifiable explanation for the observations. Case in point, the authors frequently refer to a PERCENT improvement (eg from 70 to 100% more peptides or proteins identified in these difficult samples using FAIMS, compared to a 10% drop for the more conventional sample (cultured cells).  What the authors seems to de-emphasize the fact that the FFPE and exosomal materials resulted in roughly 10 fold fewer peptides/ proteins vs the cell culture material. That in itself is a peculiar result.  Why such a drastic reduction?  Assuming one worked with roughly equal material (10ug)? Perhaps the FFPE sample can be explained - poor digestion & crosslinked peptides would be far more difficult to identify by MS. Although the gel image of Figure 2 makes it difficult to agree that the authors are consistently working with the same mass of sample across all material studied.

To move to my point, what I feel is missing from this work is an explanation or cause that justifies the observed improvement - specific to these samples (FFPE, exosomes) but not to proteins from cell culture.  What specifically is the working hypothesis?  Is it that a less complex sample, containing fewer diversity of proteins, FAIMS will improve the total detection? Yet this explanation would counter the general idea that more complex proteomic samples would benefit from more separation. Perhaps it it related to impurities in the sample (and that FAIMS is able to isolate the peptides from background, resulting in improved signal to noise)? Perhaps it is connected to the total amount of sample being loaded (assuming FFPE and perhaps also exosomes started with less material)?Perhaps if one started with fewer cells from the cell culture, to the point that the total protein IDs drop, the FAIMS approach would improve the result?  Well, if that is the case, it is certainly a testable theory.

As it stands, the result presenting in this manuscript is interesting, but incomplete.  It would therefore be difficult to make use of these findings. It appears to be implied that ALL exosome samples, and ALL FFPE samples will benefit from FAIMS, while all cell culture samples will show worse results with FAIMS.  Perhaps that's true - although I would find that extremely surprising.

Focusing on the written structure of the manuscript, I find that the entire discussion adds very little to explain the findings. Rather, the discussion repeats / summarizes the results, but does not place them in greater context. The intro is also a peculiar combination of information, first on FFPE, then on exosomes, with little to no connection between them.  It was an odd choice to study both of these samples in a single study, as there is little to connect them. 

It is my recommendation that the authors design experiments to test working theories that explain their observations.  And that the manuscript be rewritten to include these justifications.

From abstract, with similar phrasing in results section: "A slightly increase in the number of identified and quantified proteins was associated to a decrease in the number of identified and quantified peptides with FAIMS in the cell protein extract TMT experiment."  Grammatically, this sentence is okay (though replace 'was associated to a decrease...' with 'was associated with a decrease...').  However, it's just a hard sentence to wrap your head around, and would benefit from a more direct, simpler description.

Reviewer 4 Report

1, in Figure 2, the section of method and materials, for ten paired FFPE and plasma samples, 5 samples are wild-type and another 5 are mutant. But the label of 2B and 2C, there are 4 WT and 6 mutant. Also check 2F, it looks like the gel from 2F and gel from 2C are same one, based on the intensity of protein bands in each lanes. But the label of 2C and 2F are not consistent.  Similar issue in line120 – 123, the supporting information table contains 6 mutant and 4 WT, which does not align with text line 120 -123.

2, In section of Figure 4 (text section line 409 -423), the cell samples show a opposite result compared to tissue and plasma samples, because there is decreased number observed in cell sample when using FAIMS, the explanation that the author gave is not convincing. Because the increase time should be used for all three samples, while only cell sample got decreased number. The author may explain from the difference of the three samples.

3, In section 3.3. FAIMS analyses with two or three CVs, when author explains Figure 6 and makes comparison to 2 CV data, especially for Figure 6B, the author should also indicate the source/location of the 2CV data, which should be in Figure 4D and 4F.

4, in most of the result part, the author treated tissue sample as one kind and the plasma sample as another kind. But the author also used 10 paired tissue and plasma sample, if author can use a small portion to discuss the comparison of each paired tissue and plasma data, then talk about the overall result, this will be more convincing.

Overall, the English is well-written and easy to understand. The structure and storytelling are logical and persuasive. However, the author may have included a slightly excessive amount of introduction and background information in the discussion section. To enhance the clarity of the paper, the author can further explore the advantages and disadvantages of FAIMS technology.